# Vitamin D Receptor Mediates Attenuating Effect of Lithocholic Acid on Dextran Sulfate Sodium Induced Colitis in Mice

**DOI:** 10.3390/ijms24043517

**Published:** 2023-02-09

**Authors:** Hitomi Kubota, Michiyasu Ishizawa, Makoto Kodama, Yoshihiro Nagase, Shigeaki Kato, Makoto Makishima, Kenichi Sakurai

**Affiliations:** 1Division of Biochemistry, Department of Biomedical Sciences, Nihon University School of Medicine, 30-1 Oyaguchi-kamicho, Itabashi-ku, Tokyo 173-8610, Japan; 2Department of Surgery, The Nippon Dental University School of Life Dentistry, 2-3-16 Fujimi, Chiyoda-ku, Tokyo 102-8158, Japan; 3Department of Pathology, Tokyo Yamate Medical Center, 3-22-1 Hyakunin-cho, Shinjuku-ku, Tokyo 169-0073, Japan; 4Graduate School of Science and Technology, Iryo Sosei University, 5-5-1 Iino, Chuodai, Iwaki, Fukushima 970-8044, Japan; 5Research Institute of Innovative Medicine, Tokiwa Foundation, Kaminodai-57 Jobankamiyunagayamachi, Iwaki, Fukushima 972-8322, Japan

**Keywords:** vitamin D receptor, lithocholic acid, bile acid, inflammatory bowel disease, dextran sulfate sodium, intestinal mucosal injury, hypercalcemia

## Abstract

Bile acids are major components of bile; they emulsify dietary lipids for efficient digestion and absorption and act as signaling molecules that activate nuclear and membrane receptors. The vitamin D receptor (VDR) is a receptor for the active form of vitamin D and lithocholic acid (LCA), a secondary bile acid produced by the intestinal microflora. Unlike other bile acids that enter the enterohepatic circulation, LCA is poorly absorbed in the intestine. Although vitamin D signaling regulates various physiological functions, including calcium metabolism and inflammation/immunity, LCA signaling remains largely unknown. In this study, we investigated the effect of the oral administration of LCA on colitis in a mouse model using dextran sulfate sodium (DSS). Oral LCA decreased the disease activity of colitis in the early phase, which is a phenotype associated with the suppression of histological injury, such as inflammatory cell infiltration and goblet cell loss. These protective effects of LCA were abolished in VDR-deleted mice. LCA decreased the expression of inflammatory cytokine genes, but this effect was at least partly observed in VDR-deleted mice. The pharmacological effect of LCA on colitis was not associated with hypercalcemia, an adverse effect induced by vitamin D compounds. Therefore, LCA suppresses DSS-induced intestinal injury in its action as a VDR ligand.

## 1. Introduction

Bile acids are major components of bile and are secreted into the intestine in response to dietary lipids [1,2]. Bile acids emulsify dietary lipids in the intestine, a mechanism required for their digestion and absorption. In humans, primary bile acids, such as cholic acid and chenodeoxycholic acid (CDCA), are generated from cholesterol in the liver and are secreted into the bile as glycine and taurine conjugates. After assisting in lipid digestion and absorption, most bile acids enter the enterohepatic circulation and are recycled back to the liver. Bile acids that escape reabsorption are converted to secondary bile acids, such as deoxycholic acid (DCA) and lithocholic acid (LCA), by the intestinal microflora [3]. DCA and, to a much lesser extent, LCA are reabsorbed in the lower intestine. In mice, CDCA is a minor bile acid because it is metabolized to α-muricholic acid and β-muricholic acid [4]. In addition to their detergent function, bile acids act as signaling molecules that activate nuclear receptors, such as farnesoid X receptor (FXR) and vitamin D receptor (VDR), and G protein-coupled receptors, such as Takeda G protein-coupled receptor 5 (TGR5). They also regulate several physiological and cellular functions [1].

VDR is a nuclear receptor that is activated by the active form of vitamin D, 1α,25-dihydroxyvitamin D_3_ (1,25(OH)_2_D_3_), and mediates vitamin D signaling in various physiological functions, including the regulation of calcium and bone metabolism, immunity, and inflammation [5]. VDR is highly expressed not only in the small intestine but also in the large intestine, and it controls the permeability of intestinal mucosa, intestinal microflora, and immune and inflammatory responses [6]. Vitamin D deficiency is a risk factor for the exacerbation of inflammatory bowel disease (IBD) [7,8]. VDR deletion promotes colitis in mouse models [8,9], and VDR activation by vitamin D compounds suppresses inflammation [7,10]. The secondary bile acid LCA is a VDR ligand that is abundant in the colon [3]. While oral administration of 1,25(OH)_2_D_3_ activates VDR in the upper small intestine, oral LCA administration selectively activates VDR in the lower small intestine [11]. Although DCA avidly enters the enterohepatic circulation as well as primary bile acids, LCA is poorly absorbed in the intestine and is excreted in feces [12,13]. In this study, we examined whether orally administered LCA exhibits a pharmacologic action on colitis in a mouse model and determined whether this effect is mediated by VDR.

## 2. Results

### 2.1. Oral Administration of LCA Decreases Disease Activity of DSS-Induced Colitis

LCA, unlike CDCA, is poorly absorbed in the intestine [12]. Plasma bile acid levels are increased in mice fed CDCA-supplemented chow but not in mice fed LCA-supplemented chow [13], and fecal LCA levels are increased in mice after oral CDCA administration [14]. We selected an oral dose of LCA at 0.8 mmol/kg, which is not toxic but can effectively induce VDR-target gene expression in the intestine [11]. We orally administered LCA or CDCA to mice and determined whether oral administration of these bile acids could suppress DSS-induced colitis (Figure 1A). DSS treatment from day 0 to day 6 increased the disease activity index (DAI) score from day 4, reaching its peak at day 9 (Figure 1B). Symptoms of colitis improved gradually by day 15. Oral administration of LCA attenuated disease activity at days 5 and 6, although it did not decrease the peak DAI score (Figure 1B). CDCA administration had similar effects. Therefore, LCA and CDCA suppress disease activity of DSS-induced colitis at the progression stage.

Next, we examined the effect of the oral administration of LCA on DSS-induced colitis at an early stage in more detail. We orally administered LCA to mice with or without DSS treatment and performed histological analyses at day 6 (Figure 2A). DSS treatment decreased body weight (Figure 2B), shortened colon length (Figure 2C), and increased DAI scores (Figure 2D). Although LCA did not change body weight or colon length (Figure 2B,C), it effectively decreased the DAI scores in DSS-treated mice (Figure 2D). Histological evaluation showed diminished goblet cells and inflammatory cell infiltration in the intestinal mucosa in mice with DSS-induced colitis (Figure 2E). Alcian blue staining for acidic mucins also showed a lack of normal goblet cell morphology (Figure 2F). Oral LCA administration alleviated inflammatory cell infiltration and goblet cell disappearance in the colon of DSS-treated mice (Figure 2E,F). LCA administration at 0.8 mmol/kg did not induce any toxic effects in the colons of mice in the absence of DSS treatment. Thus, oral LCA treatment suppresses histological damage in DSS-induced colitis.

### 2.2. VDR Mediates the Attenuating Effect of LCA on DSS-Induced Colitis

LCA acts as a ligand not only for VDR but also for other receptors, such as FXR, pregnane X receptor (PXR), and TGR5 [15,16,17,18,19]. We determined whether VDR mediates the effect of LCA on DSS-induced colitis. We treated *Vdr(+/−)* and *Vdr(−/−)* mice with LCA by oral administration and examined the effect of LCA on DSS-induced colitis in these mice (Figure 3A). *Vdr(−/−)* mice were raised on a high-calcium and high-lactose diet to normalize blood calcium levels [20], and *Vdr(+/−)* mice were bred under the same feeding conditions (Figure 3A). DSS treatment decreased body weight and shortened colon length in both *Vdr(+/−)* and *Vdr(−/−)* mice (Figure 3B,C). Oral LCA administration did not change these phenotypes. LCA administration effectively decreased DAI scores in *Vdr(+/−)* mice (Figure 3D), a similar effect to what was observed in wild-type mice (Figure 2D). Importantly, LCA was not effective in decreasing DAI scores in *Vdr(−/−)* colitis mice. Histological examination revealed severe colitis, including erosion, the disappearance of intestinal crypts and goblet cells, epithelial hyperplasia, and inflammatory cell infiltration in the lamina propria, in both *Vdr(+/−)* and *Vdr(−/−)* mice (Figure 3E). While LCA apparently suppressed histological damage of colitis in *Vdr(+/−)* mice, these effects were abolished in *Vdr(−/−)* mice (Figure 3E).

We next examined the effect of LCA on the expression of genes related to inflammation. As previously reported [21,22,23], the mRNA levels of inflammatory cytokines, *Il6* (the gene encoding interleukin 6 (IL-6), *Il17a*, *Tnf* (the gene encoding tumor necrosis factor α), and *Il1b*, were increased in the colon of *Vdr(+/−)* mice with DSS-induced colitis (Figure 4A–D). Interestingly, compared with those of *Vdr(+/−)* mice, *Il6* mRNA levels were highly elevated in *Vdr(−/−)* mice with DSS treatment (Figure 4A). LCA administration decreased *Il6* expression in *Vdr(+/−)* mice, but its effect was not significant in *Vdr(−/−)* mice. LCA treatment lowered the mRNA levels of *Il17a* and *Tnf* and tended to decrease *Il1b* expression in *Vdr(−/−)* mice (Figure 4B–D). These effects were similar to those in *Vdr(+/−)* mice, although the effect of LCA was not statistically significant for *Il17a* expression. These findings indicate that LCA attenuates DSS-induced colitis in a VDR-dependent manner and suppresses inflammatory cytokine expression at least in part by VDR-independent mechanism(s).

Among the intestinal epithelial tight junction proteins, claudin-15 plays an important role in VDR-mediated barrier function as its gene, *Cldn15*, is a VDR target [24]. We examined the *Cldn15* expression in the colon of DSS-treated mice. Consistent with a previous report [24], *Cldn15* mRNA levels were lower in the colon of *Vdr(−/−)* mice (Figure 5). LCA treatment showed a tendency to increase *Cldn15* expression in *Vdr(+/−)* mice, but it was not effective in *Vdr(−/−)* mice. *Cldn15* mRNA levels were significantly lower in LCA-administered *Vdr(−/−)* mice than in *Vdr(+/−)* mice.

### 2.3. Oral Dosing of LCA Does Not Increase Plasma Calcium and Aminotransferase Levels

The active form of vitamin D, 1,25(OH)_2_D_3_, plays an important role in calcium homeostasis in the body as a potent VDR ligand [25]. Vitamin D deficiency causes hypocalcemia, and the administration of 1,25(OH)_2_D_3_ and its analogs increases serum calcium levels. However, the hypercalcemic activity becomes a problematic adverse effect in the clinical application of VDR ligands to calcium-metabolism-unrelated diseases, including malignancies and inflammatory diseases [26,27,28,29]. We previously demonstrated that plasma calcium levels are increased in mice treated with 1,25(OH)_2_D_3_ at an oral dose that induces the ileal expression of the VDR target gene *Cyp24A1* to levels similar to those by LCA (0.8 mmol/kg) [11]. Daily dosing of LCA for 8 days did not increase plasma calcium levels in *Vdr(+/−)* or *Vdr(−/−)* mice fed a high-calcium diet (Figure 6A). Although LCA is known to induce cholestasis in rodents [17,18], LCA administration did not increase aspartate aminotransferase or alanine aminotransferase levels under our experimental conditions (Figure 6B). Therefore, LCA can exert an inhibitory effect on DSS-induced colitis via VDR without inducing hypercalcemia or hepatotoxicity.

## 3. Discussion

In this study, we found that the oral administration of LCA attenuates DSS-induced colitis in a VDR-dependent manner. VDR is a nuclear receptor that is activated by 1,25(OH)_2_D_3_ and mediates vitamin D signaling [5]. Bile acids act not only as detergents for the intestinal digestion and absorption of lipid-soluble nutrients but also as signaling molecules that activate nuclear and membrane receptors [1]. The secondary bile acid LCA is generated from the primary bile acid CDCA by the intestinal microflora [3] and is another natural VDR ligand [16]. Unlike other bile acids, most of which are reabsorbed in the intestine and are recycled back to the liver, LCA is poorly absorbed and does not normally accumulate in the enterohepatic circulation [12]. We previously showed that oral LCA administration does not increase plasma levels of total bile acids or LCA [13]. Intraperitoneal administration of LCA and ursodeoxycholic acid has been reported to suppress intestinal inflammation in DSS-treated mice [21,30]. However, ursodeoxycholic acid is not a VDR ligand [16], and these studies have not shown what receptor(s) mediate the pharmacological action of LCA and ursodeoxycholic acid [21,30]. Thus, our study is the first to demonstrate the VDR-dependent effect of LCA in the intestinal lumen.

Vitamin D signaling via VDR plays a protective role in experimental colitis [31]. 1,25(OH)_2_D_3_ treatment suppresses colitis in IL-10 knockout mice and DSS-treated mice [32,33]. Epithelial VDR signaling plays an important role in the intestinal mucosal barrier and in protection against colitis [9,34,35]. VDR function in Paneth cells is necessary for antibacterial activities and protection against intestinal injury [36]. The DAI scores at day 6 were 4.5, 8.8, and 10.2 for DSS-treated wild-type, *Vdr(+/−)*, and *Vdr(−/−)* mice, respectively (Figure 2D and Figure 3D), which is consistent with the previous reports that *Vdr(−/−)* mice have more severe IBD symptoms than wild-type mice [33,34,37]. These findings suggest that VDR expression levels influence the severity of colitis, although confounding factors associated with breeding conditions may affect disease activity. The oral administration of LCA suppressed disease activity during the early phase (Figure 1) and protected the colon against mucosal injury in DSS-treated mice (Figure 2). These effects of LCA were dependent on the presence of VDR (Figure 3). The protective effect of intestinal VDR against colitis is associated with expression of the VDR target gene *Cldn15*, which encodes the tight junction protein claudin-15 [24]. *Cldn15* expression was lower in LCA-administered *Vdr(−/−)* mice than in *Vdr(+/−)* mice (Figure 5). These results support the protective role of the LCA–VDR axis in the intestinal mucosal barrier. LCA treatment did not protect body weight loss and shortening of the colon nor did it decrease the peak of disease activity in DSS-treated mice (Figure 1 and Figure 2). Thus, the effect of LCA is partially effective in the early phase of injury in DSS-induced colitis. This is likely due to the poor absorption of LCA into the circulation. VDR in immune cells, such as macrophages, also mediates vitamin D signaling in the suppression of inflammation [38,39]. VDR is involved in regulatory T-cell homeostasis by responding to microbial bile acid metabolites [40]. VDR in immune cells may also be activated by LCA or its metabolites and be involved in the anti-inflammatory effect.

Although VDR deletion abolished the protective effect of LCA on DAI scores and histological injury in DSS-induced colitis (Figure 3), LCA administration decreased the mRNA expression of inflammatory cytokines in both *Vdr(+/−)* and *Vdr(−/−)* mice, although to differing extents (Figure 4). LCA is a ligand not only for VDR but also for other bile acid receptors, such as FXR, PXR, and TGR5 [1]. Because the activation of FXR, PXR and TGR5 can suppress DSS-induced colitis [41,42,43,44], the effect of LCA on inflammatory cytokine expression may be mediated by these receptors. Because VDR is necessary for the effect of LCA on DAI scores and histological injury (Figure 3), VDR-mediated LCA signaling plays a main role in the protection against colitis.

The oral administration of CDCA also showed an attenuating effect on the disease activity of DSS-induced colitis (Figure 1). CDCA was effective in an early phase of colitis, like LCA. CDCA is converted to LCA by the intestinal microflora [3], and fecal LCA levels are increased by oral CDCA administration [14]. The effect of oral CDCA administration likely occurs through conversion to LCA and VDR activation. However, CDCA is a potent FXR agonist [15], and FXR activation suppresses DSS-induced colitis [41,44]. In addition to FXR agonist activity, CDCA activates the NLRP3 inflammasome and exacerbates hepatic inflammation in mice [45]. It is possible that CDCA attenuates colitis through a combination of several mechanisms.

Vitamin D insufficiency and deficiency are associated with an increased risk of IBD in humans, and polymorphisms in the *VDR* gene are associated with IBD susceptibility [7,8]. Patients with ulcerative colitis have decreased levels of LCA and DCA in the stool and a decreased proportion of secondary bile-acid-producing intestinal bacteria [46,47]. These findings suggest a physiological role of the LCA–VDR axis in intestinal homeostasis, contributing to the protection against IBD. Based on epidemiological and experimental evidence, thousands of vitamin D analogs have been developed for the treatment of bone and mineral disorders, cancer, autoimmune and inflammatory diseases, infection, and cardiovascular disease, but adverse effects, particularly hypercalcemia, limit clinical application [48,49]. LCA exerted a protective effect against DSS-induced colitis without increasing plasma calcium levels (Figure 6). LCA derivatives, LCA acetate and LCA propionate, are VDR activators with less or no calcemic activity [50]. Thus, LCA derivatives may have potential for IBD therapy [51].

## 4. Materials and Methods

### 4.1. Animals and Treatment

Seven-week-old male C57BL/6J mice were purchased from CLEA Japan (Tokyo, Japan). Mice were raised on a standard diet (CE-2; CLEA Japan) in a specific pathogen-free facility. VDR-null (*Vdr(−/−)*) mice and heterozygous (*Vdr(+/−)*) mice were obtained by breeding *Vdr(+/−)* mice on a pure C57BL/6J background [52]. Original *Vdr(−/−)* mice were generated by Dr. Kato’s laboratory [20] and were backcrossed with C57BL/6J mice for at least ten generations. *Vdr(+/−)* and *Vdr(−/−)* male mice were raised on a high-calcium and high-lactose diet to normalize blood calcium levels in *Vdr(−/−)* mice [20] and used for experiments at 8 to 12 weeks of age. All mice were maintained under controlled temperature (23 ± 1 °C) and humidity (45–65%) with free access to water. Mice were randomly divided into the following groups: DSS, DSS + LCA and DSS + CDCA (*n* = 6 for each group), or control, LCA, DSS, and DSS + LCA (*n* = 6–10 for each group). Mice were orally treated with control corn oil, LCA (0.8 mmol/kg; Nacalai Tesque, Kyoto, Japan), or CDCA (0.8 mmol/kg; Nacalai Tesque) from day −2 to day 14 once a day (Figure 1A), or from day −2 to day 5 once a day (Figure 2A). LCA and CDCA were dissolved in corn oil. For the DSS-induced colitis model, mice received 2% or 3% DSS (MP Biomedicals, Santa Ana, CA, USA) dissolved in water for 6 days. The disease activity of colitis was evaluated by scoring weight loss compared with initial weight, stool consistency, and bleeding [53] (Table 1). Body weight and DAI scores were monitored daily [22]. Mice were euthanized with carbon dioxide for analysis of blood and tissue samples, as shown in Figure 1A and Figure 2A. The plasma calcium concentrations and aminotransferase levels were quantified with a Calcium C Test wako and transaminase CII Test wako (Fujifilm Wako Pure Chemical Corporation, Osaka, Japan), respectively [50,54]. All animal experiments were performed according to protocols that adhered to the Nihon University Animal Care and Use Committee and conformed to the ARRIVE guidelines.

### 4.2. Histology

Distal colon samples were fixed with 4% paraformaldehyde (Fujifilm Wako Pure Chemical Corporation, Richmond, VA, USA) for 24 h. Then, tissues were embedded in paraffin and cut into 3 μm sections. After deparaffinization with graded concentrations of xylene and ethanol, sections were stained with hematoxylin and eosin and with Alcian blue. Stained specimens were examined by a single pathologist in blinded fashion, as previously described [22,35]. Histological scores were defined as follows: 0, none; 1, slight; 2, moderate; 3, severe. Each score was multiplied by a percentage of the extent of the lesion (×1, 0–25%; ×2, 26–50%; ×3, 51–75%; ×4, 76–100%) and used for the assessment.

### 4.3. Reverse-Transcription and Quantitative Real-Time Polymerase Chain Reaction

Total RNA was isolated from tissue samples using the acid guanidium thiocyanate/phenol/chloroform method and was purified with lithium chloride, as previously reported [11,22]. cDNAs were synthesized using a ImProm Ⅱ Reverse-Transcription system (Promega Corporation, Madison, WI, USA), and real-time polymerase chain reactions were performed on a StepOne Plus Real-time PCR System (Thermo Fisher Scientific, Waltham, MA, USA) using Power SYBR Green PCR Master Mix (Thermo Fisher Scientific) [11]. Primer sequences for mouse *Cdn15* were 5′-TCT TTC TAG GCA TGG TGG GA-3′ and 5′-TCA GTA GTG ATG TTG ACG GC-3′, and those for mouse *Il6*, *Il17a*, *Tnf*, *Il1b*, and *Rn18s* (the gene encoding 18S ribosomal RNA) were previously reported [22,54]. The mRNA values were normalized to the *Rn18s* levels.

### 4.4. Statistical Analysis

Data are presented as means ± S.D. One-way ANOVA followed by Tukey’s multiple comparisons was performed to analyze the data of more than two groups, unpaired two-tailed Student’s *t*-test was used to compare two groups, and two-way ANOVA was used to analyze the influence of two different factors using Prism 8 (GraphPad Software, La Jolla, CA, USA).

## Figures and Tables

**Figure 1 ijms-24-03517-f001:**
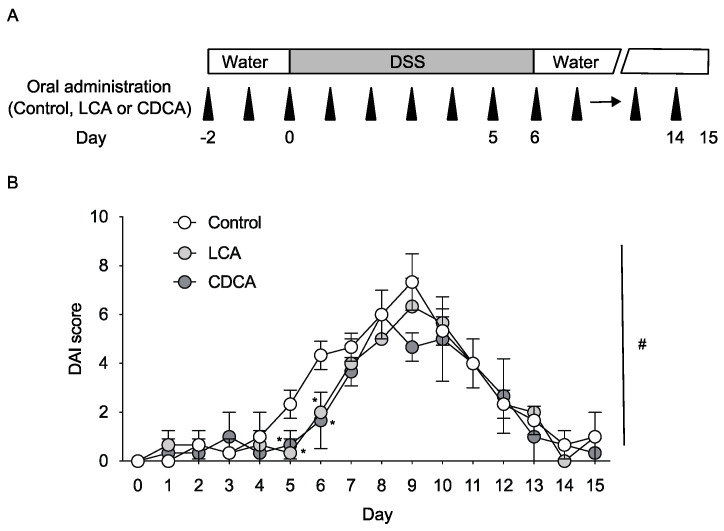
Effects of oral administration of LCA and CDCA on symptoms of colitis in DSS-treated mice. (**A**) Experimental procedure for oral administration of LCA and CDCA and DSS treatment in mice. Mice were orally administered control corn oil, LCA (in corn oil), or CDCA (in corn oil) from day −2 to day 14. To generate colitis, mice received 3% DSS from day 0 to 6. *n* = 6 for each group. (**B**) Time course of DAI scores from day 0 to 15. ^#^
*p* < 0.05 (two-way ANOVA). Data are presented as means ± S.D. * *p* < 0.05 compared with control (one-way ANOVA followed by Tukey’s multiple comparisons).

**Figure 2 ijms-24-03517-f002:**
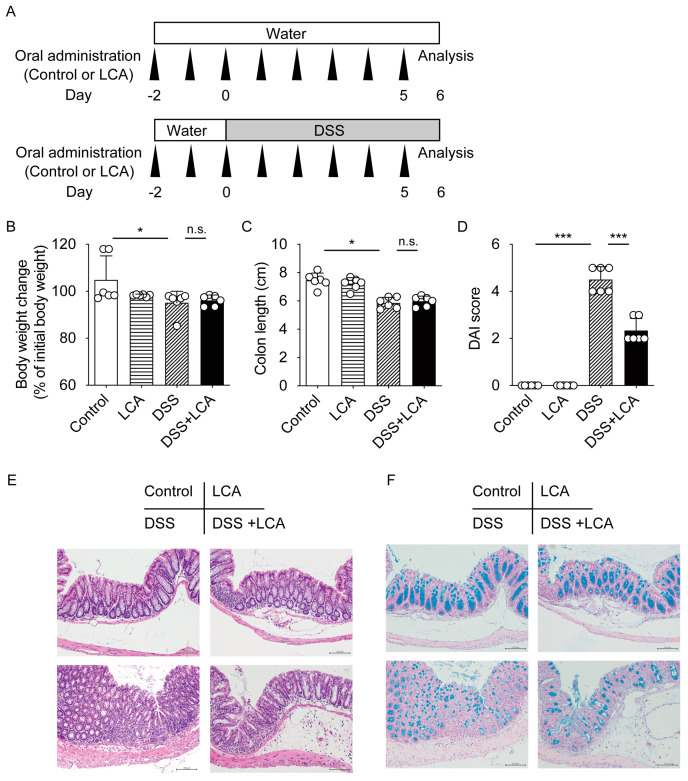
Oral administration of LCA suppresses histological injury in an early phase of DSS-induced colitis. (**A**) Experimental procedure for the assessment of the effect of oral LCA in an early phase of DSS-induced colitis. Mice were subjected to analysis at day 6. *n* = 6 for each group. (**B**) Body weight change. (**C**) Colon length. (**D**) DAI score. Data are presented as means ± S.D. * *p* < 0.05; *** *p* < 0.001; n.s., not significant (one-way ANOVA followed by Tukey’s multiple comparisons). (**E**) Histology with hematoxylin and eosin staining. (**F**) Histology with Alcian blue staining. Scale bar, 100 μm. Original magnification 200×.

**Figure 3 ijms-24-03517-f003:**
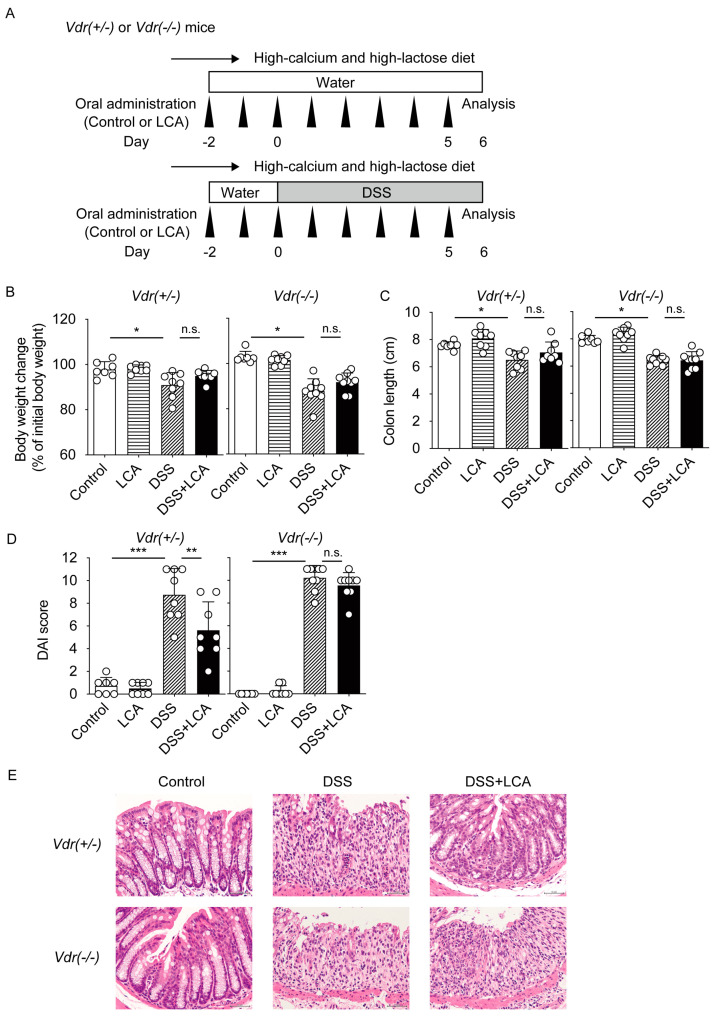
Effects of oral administration of LCA on DSS-induced colitis in *Vdr(+/−)* and *Vdr(−/−)* mice. (**A**) Experimental procedure for oral administration of LCA and DSS treatment in *Vdr(+/−)* and *Vdr(−/−)* mice. Mice were raised on a high-calcium and high-lactose diet. Mice were administered control corn oil or LCA (in corn oil) from day −2 to day 5, were treated with or without 2% DSS from day 0 to 6, and were subjected to analysis at day 6. *n* = 8–10 for each group. (**B**) Body weight change. (**C**) Colon length. (**D**) DAI score. Data are presented as means ± S.D. * *p* < 0.05; ** *p* < 0.01; *** *p* < 0.001; n.s., not significant (one-way ANOVA followed by Tukey’s multiple comparisons). (**E**) Histology with hematoxylin and eosin staining. Scale bar, 50 μm. Original magnification 400×.

**Figure 4 ijms-24-03517-f004:**
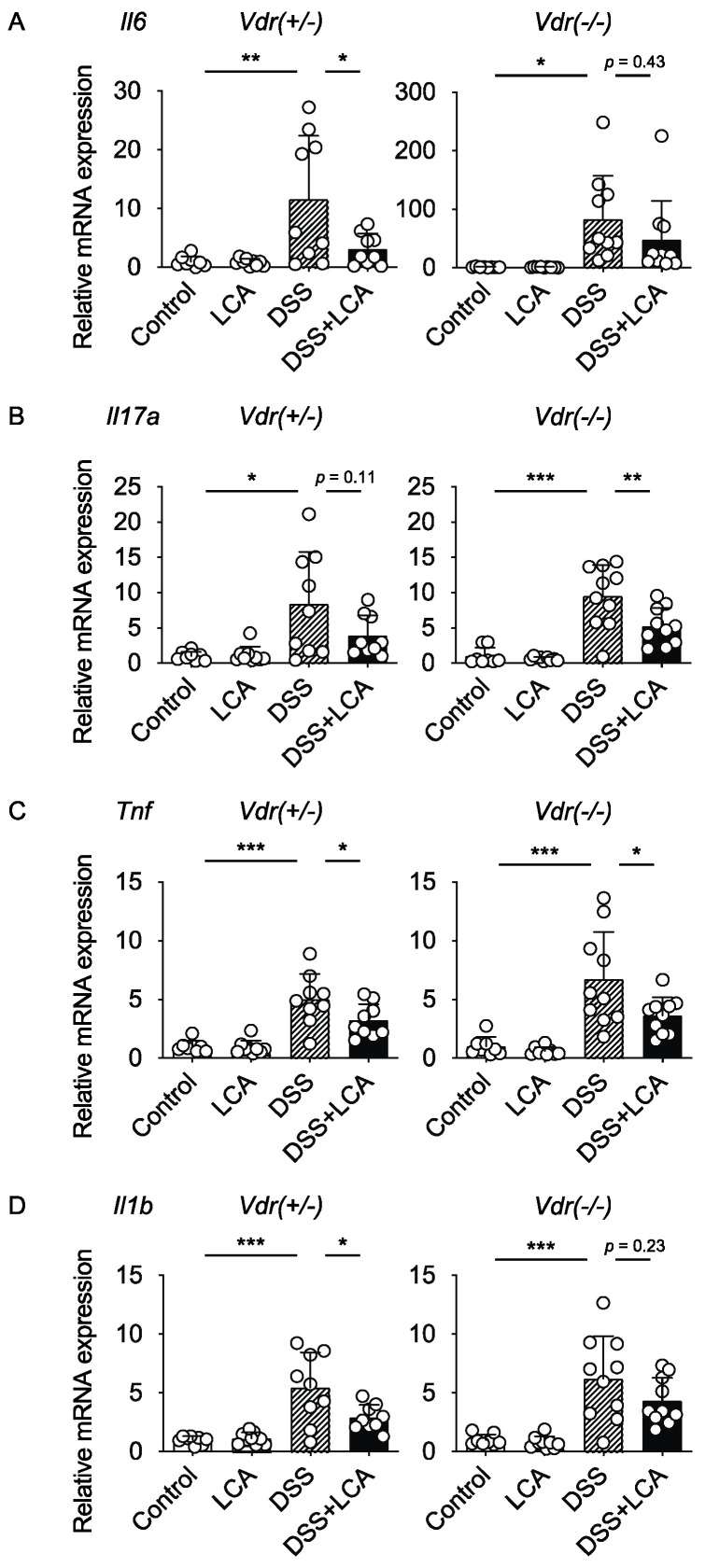
Effect of oral administration of LCA on inflammatory cytokine expression in the colon of *Vdr(+/−)* and *Vdr(−/−)* mice. Colon samples were subjected to analysis at day 6, as shown in Figure 3A, and mRNA expressions of (**A**) *Il6*, (**B**) *Il17a*, (**C**) *Tnf*, and (**D**) *Il1b* were determined with reverse-transcription and quantitative real-time polymerase chain reaction. Values for normalized mRNA expression are relative to those of control *Vdr(+/−)* mice. Data are presented as means ± S.D. * *p* < 0.05; ** *p* < 0.01; *** *p* < 0.001 (one-way ANOVA followed by Tukey’s multiple comparisons).

**Figure 5 ijms-24-03517-f005:**
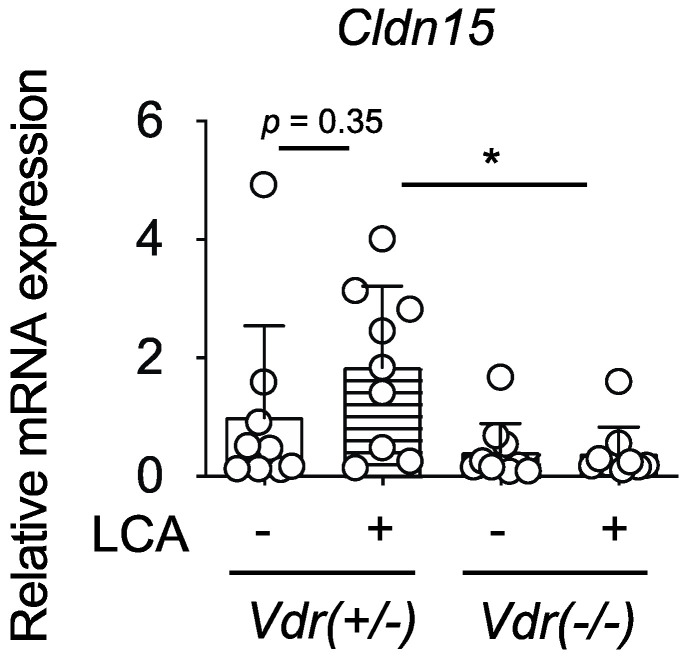
Effect of oral administration of LCA on expression of *Cldn15* in the colon of DSS-treated *Vdr(+/−)* and *Vdr(−/−)* mice. Colon samples were subjected to analysis at day 6, as shown in Figure 3A, and mRNA expression of *Cldn15* was determined with reverse-transcription and quantitative real-time polymerase chain reaction. Values for normalized mRNA expression are relative to those of *Vdr(+/−)* mice without LCA administration. Data are presented as means ± S.D. * *p* < 0.05 (one-way ANOVA followed by Tukey’s multiple comparisons).

**Figure 6 ijms-24-03517-f006:**
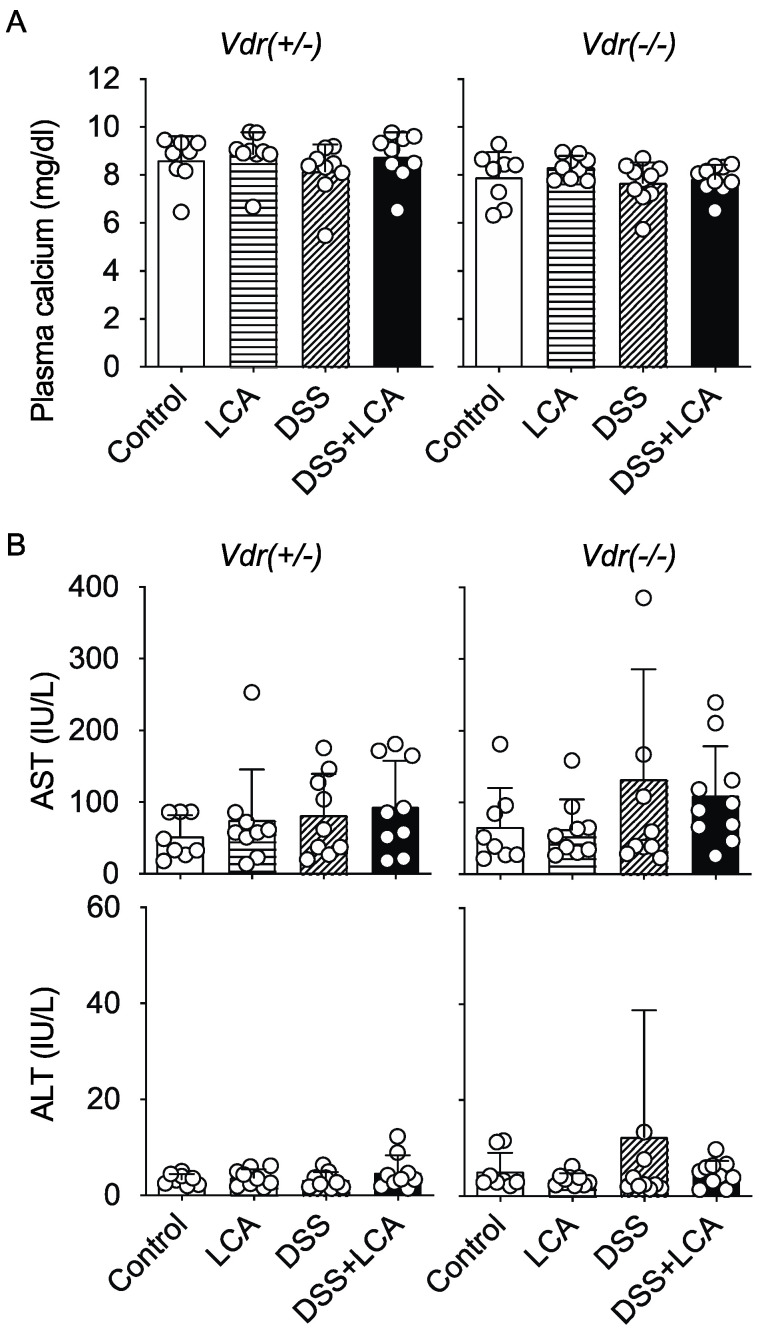
Oral LCA administration does not increase plasma calcium, aspartate aminotransferase, or alanine aminotransferase levels. (**A**) Plasma calcium levels. (**B**) Plasma levels of aspartate aminotransferase (AST) and alanine aminotransferase (ALT). Blood samples were collected at day 6, as shown in Figure 3A. Data are presented as means ± S.D.

**Table 1 ijms-24-03517-t001:** DAI score.

Item	Standard	Score
Weight loss	0	0
	1–5%	1
	6–10%	2
	11–20%	3
	>20%	4
Stool consistency	Normal	0
	Loose stool	2
	Diarrhea	4
Rectal bleeding	No blood	0
	Hemoccult positive	1
	Hemoccult positive and visual pellet bleeding	2
	Gloss bleeding, blood around anus	4

The DAI score is the sum of scores of the above items [53].

## Data Availability

The data presented in this study are available on request from the corresponding author.

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
