# Peer review of "Vitamin D Receptor Mediates Attenuating Effect of Lithocholic Acid on Dextran Sulfate Sodium Induced Colitis in Mice"

_ijms, 2023, doi:10.3390/ijms24043517_

Round 1

Reviewer 1 Report

The vitamin D receptor (VDR) is a receptor for the active form of vitamin D and lithocholic acid (LCA), which is a secondary bile acid generated from the primary bile acid CDCA by intestinal microflora. In this study, the authors explored the effects of oral administration of LCA on colitis in a DSS mice model. Treatment with LCA was shown to suppresses histological injury in an early phase of DSS-induced colitis in wild type and Vdr(+/-) mice but these were not observed in VDR-deleted mice. LCA suppresses inflammatory cytokine expression at least in part by VDR-independent mechanism. Moreover, Cldn15 mRNA levels, a VDR target, were significantly lower in LCA-administered Vdr(-/-) mice compared to Vdr(+/-) mice and oral administration of LCA did not raise serum calcium level. The authors reported a physiological role of the LCA-VDR axis in intestinal homeostasis and suggested LCA derivatives as promising IBD therapies. The major conclusions are supported by experimental results.

Major comments: 

1) Oral administration of LCA has also been used as model of cholestatic liver disease. The authors should analyze the cholestatic phenotype (histology, ALT, AST) in their LCA+DSS model to exclude the possibility that the intestinal effect might be secondary to the liver effect. 

2) LCA is toxic to the liver, the authors should be very careful in making the statement that LCA or its derivatives are “promising IBD therapies,” until they can demonstrate that these derivatives remain effective in activating VDR but not toxic.

3) The authors should provide more literature support their statement that “…problematic adverse effect in the clinical application of VDR ligands to calcium metabolism-unrelated diseases, including inflammatory disease” as stated in 2.3. 

Minor comments: 

Spelling mistake for the word “compared” in line 160.

In Figure 1, “To generate colitis, mice were received 3% DSS….” should be changed to “ To generate colitis, mice received 3% DSS….”

Spelling mistake for the word “administration” in Figures 1, 2 and 3 and “length” in Figure 2 and 3

Magnification of histological images should also be stated in Figure 2 and 3. 

In Figure 3, “Effect of oral administration….” should be changed to “Effects of oral administration….”

In Figure 4, “…mRNA expression of Il6 (A), Il17a (B), Tnf (C), and Il1b (D)….” should be changed to “…mRNA expression of (A) Il6, (B) Il17a, (C)Tnf, and (D) Il1b….”

In Figure 5, the explanation on Cldn15 is not needed in the legend, i.e., “Effect of oral administration of LCA on expression of Cldn15, a gene involved in the intestinal tight junction, in the colon of DSS-treated Vdr(+/-) and Vdr(-/-) mice.”

The style for bar graph should be consistent. Individual values should be added to Figure 6. 

In section 4.4, “We performed one-way ANOVA….” should be changed to passive voice to be consistent with the overall tone in Materials and Methods. 

Author Response

Comments by Reviewer #1

The vitamin D receptor (VDR) is a receptor for the active form of vitamin D and lithocholic acid (LCA), which is a secondary bile acid generated from the primary bile acid CDCA by intestinal microflora. In this study, the authors explored the effects of oral administration of LCA on colitis in a DSS mice model. Treatment with LCA was shown to suppresses histological injury in an early phase of DSS-induced colitis in wild type and Vdr(+/-) mice but these were not observed in VDR-deleted mice. LCA suppresses inflammatory cytokine expression at least in part by VDR-independent mechanism. Moreover, Cldn15 mRNA levels, a VDR target, were significantly lower in LCA-administered Vdr(-/-) mice compared to Vdr(+/-) mice and oral administration of LCA did not raise serum calcium level. The authors reported a physiological role of the LCA-VDR axis in intestinal homeostasis and suggested LCA derivatives as promising IBD therapies. The major conclusions are supported by experimental results.

Major comment #1

Oral administration of LCA has also been used as model of cholestatic liver disease. The authors should analyze the cholestatic phenotype (histology, ALT, AST) in their LCA+DSS model to exclude the possibility that the intestinal effect might be secondary to the liver effect.

Response

We have investigated the effect of LCA and LCA derivatives in mice (Adachi et al. J Lipid Res 46: 46, 2005; Ishizawa et al., J Lipid Res 49: 763, 2008; Nishida et al. Drug Metab Disp 37: 2037, 2009; Ishizawa et al. Int J Mol Sci 19: 1975, 2018). According to our experiments regarding to the previous paper (Ishizawa et al. Int J Mol Sci 19: 1975, 2018), we selected an oral dose of LCA, which is not toxic but can effectively induce VDR-target gene expression (lines 73-74). We examined plasma aminotransferase levels and confirmed that LCA administration did not increase these levels. We added the data to new Figure 6B and explained the results (lines 189-193).

Because LCA administration did not increase aminotransferase levels, we did not perform histological analysis in this study. We are also investigating the role of VDR in hepatic immunity (Umeda et al., J Leu Biol 106: 791, 2019). The effect of toxic doses of LCA and the role of VDR in the pathogenesis of liver diseases are under investigation. When we observe liver dysfunction, we would like to evaluate liver histology in detail.

Major comment #2

LCA is toxic to the liver, the authors should be very careful in making the statement that LCA or its derivatives are “promising IBD therapies,” until they can demonstrate that these derivatives remain effective in activating VDR but not toxic.

Response

According to the reviewer’s indication, we changed “LCA derivatives hold promise as IBD therapies” to “LCA derivatives may have potential for IBD therapy” (lines 291-292).

Major comment #3

The authors should provide more literature support their statement that “…problematic adverse effect in the clinical application of VDR ligands to calcium metabolism-unrelated diseases, including inflammatory disease” as stated in 2.3.

Response

We added references, which refer to preclinical and clinical studies of vitamin D compounds for malignancies and inflammatory diseases (lines 184-185; new ref [27]-[29]).

Minor comment #1

Spelling mistake for the word “compared” in line 160.

Response

We thank the reviewer for indicating the typographical error. We corrected it (line 171).

Minor comment #2

In Figure 1, “To generate colitis, mice were received 3% DSS….” should be changed to “ To generate colitis, mice received 3% DSS….”

Response

We thank the reviewer for indicating the English error. We corrected it (line 86).

Minor comment #3

Spelling mistake for the word “administration” in Figures 1, 2 and 3 and “length” in Figure 2 and 3

Response

We thank the reviewer for indicating the typographical errors. We corrected them in Figures 1-3.

Minor comment #4

Magnification of histological images should also be stated in Figure 2 and 3.

Response

According to the reviewer’s indication, we added information of magnification in legends to Figure 2 (line 112) and Figure 3 (139). We found the mistake in the information of scale bar in Figure 3 legend and corrected it (line 139).

Minor comment #5

In Figure 3, “Effect of oral administration….” should be changed to “Effects of oral administration….”

Response

According to the reviewer’s indication, we changed “Effect” to “Effects” (line 132).

Minor comment #6

In Figure 4, “…mRNA expression of Il6 (A), Il17a (B), Tnf (C), and Il1b (D)….” should be changed to “…mRNA expression of (A) Il6, (B) Il17a, (C)Tnf, and (D) Il1b….”

Response

According to the reviewer’s indication, we changed the indicated points (line 157).

Minor comment #7

In Figure 5, the explanation on Cldn15 is not needed in the legend, i.e., “Effect of oral administration of LCA on expression of Cldn15, a gene involved in the intestinal tight junction, in the colon of DSS-treated Vdr(+/-) and Vdr(-/-) mice.”

Response

According to the reviewer’s indication, we deleted “a gene involved in the intestinal tight junction” (line 173).

Minor comment #8

The style for bar graph should be consistent. Individual values should be added to Figure 6.

Response

According to the reviewer’s indication, we modified Figure 6.

Minor comment #9

In section 4.4, “We performed one-way ANOVA….” should be changed to passive voice to be consistent with the overall tone in Materials and Methods.

Response

According to the reviewer’s indication, we modified the sentence (lines 354-356).

Reviewer 2 Report

The paper by Kubota et al. addresses the problem of the attenuating role of the lithocholic acid (LCA) in colitis in mice. The fact that LCA attenuates the severity of intestine inflammation has been reported before. In this work, the authors tested orally administered LCA, and they asked about the role of vitamin D receptor (VDR) in the actions of LCA. VDR is the intracellular receptor for the active form of vitamin D, but it has been documented that LCA is a week agonist of this receptor as well.

The paper has been well designed, and well written, however there are some minor issues that need to be addressed before publication.

1. Disease activity index (DAI) must be explained in a detailed way. The best would be the table with description of every score.

2. In every graph the meaning of the whiskers must be explained.

3. Gene names should be written using italics.

4. Line 86: the text should be “mice received” instead of “mice were received”.

5. In wild type mice DSS-induced colitis reached DAI score of about 5 (Figure 2D), while in Vdr(-/+) DSS- induced colitis was up to 9, and in Vdr(-/-) mice it reached DAI score of 11 (Figure 3D). Please, explain why.

Author Response

Comments by Reviewer #2

The paper by Kubota et al. addresses the problem of the attenuating role of the lithocholic acid (LCA) in colitis in mice. The fact that LCA attenuates the severity of intestine inflammation has been reported before. In this work, the authors tested orally administered LCA, and they asked about the role of vitamin D receptor (VDR) in the actions of LCA. VDR is the intracellular receptor for the active form of vitamin D, but it has been documented that LCA is a week agonist of this receptor as well.

The paper has been well designed, and well written, however there are some minor issues that need to be addressed before publication.

Comment #1

Disease activity index (DAI) must be explained in a detailed way. The best would be the table with description of every score.

Response

According to the reviewer’s indication, we added the explanation of DAI score with adding a new table (Table 1) (lines 310-311).

Comment #2

In every graph the meaning of the whiskers must be explained.

Response

According to the reviewer’s indication, we added the information (lines 87-88 in Figure 1; line 109 in Figure 2; line 137 in Figure 3; line 159 in Figure 4; line 177 in Figure 5; line 202 in Figure 6).

Comment #3

Gene names should be written using italics.

Response

We thank the reviewer for indicating this problem. I found that some italics were converted to non-italics after I uploaded the manuscript file to the journal site. We corrected them.

Comment #4

Line 86: the text should be “mice received” instead of “mice were received”.

Response

We thank the reviewer for indicating the English error. We corrected it (line 86).

Comment #5

In wild type mice DSS-induced colitis reached DAI score of about 5 (Figure 2D), while in Vdr(-/+) DSS- induced colitis was up to 9, and in Vdr(-/-) mice it reached DAI score of 11 (Figure 3D). Please, explain why.

Response

We thank the reviewer for checking the manuscript carefully. We added the explanation as follows:

(lines 224-229)

“The DAI scores at day 6 were 4.5, 8.8, and 10.2 for DSS-treated wild-type, Vdr(+/-), and Vdr(-/-) mice, respectively (Figure 2D and Figure 3D), consistent with the previous reports that Vdr(-/-) mice have more severe IBD symptoms than wild-type mice [33,34,37]. These findings suggest VDR expression levels influence the severity of colitis, although confounding factors associated with breeding conditions may affect disease activity.”